# Development of Group B Coxsackievirus as an Oncolytic Virus: Opportunities and Challenges

**DOI:** 10.3390/v13061082

**Published:** 2021-06-05

**Authors:** Huitao Liu, Honglin Luo

**Affiliations:** 1Centre for Heart Lung Innovation, St. Paul’s Hospital—University of British Columbia, Vancouver, BC V6Z 1Y6, Canada; Huitao.liu@hli.ubc.ca; 2Department of Experimental Medicine, University of British Columbia, Vancouver, BC V6Z 1Y6, Canada; 3Department of Pathology and Laboratory Medicine, University of British Columbia, Vancouver, BC V6Z 1Y6, Canada

**Keywords:** coxsackievirus B, oncolytic virus, cancer, virotherapy, cancer, toxicity, anti-tumor immunity

## Abstract

Oncolytic viruses have emerged as a promising strategy for cancer therapy due to their dual ability to selectively infect and lyse tumor cells and to induce systemic anti-tumor immunity. Among various candidate viruses, coxsackievirus group B (CVBs) have attracted increasing attention in recent years. CVBs are a group of small, non-enveloped, single-stranded, positive-sense RNA viruses, belonging to species *human Enterovirus B* in the genus *Enterovirus* of the family *Picornaviridae.* Preclinical studies have demonstrated potent anti-tumor activities for CVBs, particularly type 3, against multiple cancer types, including lung, breast, and colorectal cancer. Various approaches have been proposed or applied to enhance the safety and specificity of CVBs towards tumor cells and to further increase their anti-tumor efficacy. This review summarizes current knowledge and strategies for developing CVBs as oncolytic viruses for cancer virotherapy. The challenges arising from these studies and future prospects are also discussed in this review.

## 1. Introduction

Oncolytic virotherapy represents a new and promising option for cancer treatment. This approach uses engineered or naturally occurring viruses, termed oncolytic viruses, to target and destroy tumor cells while sparing normal cells [1,2]. Current evidence supports that both direct lysis of cancer cells and anti-tumor immunity triggered by viral infection contribute to the anti-tumor efficacy of virotherapy [1,2]. Oncolytic viruses selectively infect and replicate in tumor cells, inducing cell destruction and the consequent release and spread of progeny virions to infect adjacent cells. Oncolytic viruses can also stimulate immunogenic cell death through promoting the innate and adaptive immune response. The advantage of virotherapy over other classical cancer therapies (e.g., chemo- and radiotherapy) lies in the fact that candidate viruses can be genetically manipulated to increase their potency against specific cancer types.

As a milestone in virotherapy, in 2015, the US FDA approved the first oncolytic agent Imlygic (Talimogene laherparepvec, T-Vec) for the treatment of melanoma [3,4]. T-Vec is a modified herpes simplex virus-1 (HSV-1) genetically engineered to encode the immunostimulatory cytokine granulocyte-macrophage colony-stimulating factor (GM-CSF). Although the survival benefits remain controversial, the approval of T-Vec has promoted significant interest in the development of oncolytic viruses as either a monotherapy or in combination with other anti-tumor therapies [3,4]. According to clinicaltrials.gov, there are currently 105 registered oncolytic viral trials in the USA, 21 in Canada, 18 in Spain, and 15 in France among other top countries (Figure 1A). In recent years, the number of initiated clinical studies on virotherapy has increased significantly (Figure 1B).

Research in the field of oncolytic RNA viruses has grown rapidly over the past decade. Several RNA viruses that exhibit oncolytic activity have been tested in early clinical trials, which include Newcastle disease virus [5], measles virus [6], vesicular stomatitis virus (VSV) [7], Seneca valley virus [8], reovirus [9], respiratory syncytial virus [10], poliovirus [11], and coxsackievirus [12]. Among enteroviruses, the PVSRIPO, a modified poliovirus carrying an internal ribosomal entry site (IRES) of human rhinovirus type 2 [11,13,14], and the CVA21-based CAVATAK [15,16,17,18,19] are the two best-characterized oncolytic viruses. A number of clinical trials have been conducted on them and shown promising efficacy. In 2018, the CAVATAK start-up company, Viralytics, was acquired by Merck for $394 million. This remarkable news has been very encouraging, especially to those interested in developing enteroviruses for cancer therapy. While PVSRIPO and CAVATAK have demonstrated effectiveness against certain tumors, they have less efficacy against other tumors and can sometimes cause adverse side effects. Thus, investigation into additional enteroviruses is needed.

Coxsackieviruses are a group of non-enveloped, positive-sense, single-stranded RNA viruses belonging to the species human *Enterovirus* in the family of *Picornaviridae* [20]. Coxsackieviruses were originally classified into two groups—A (CVA) and B (CVB)—on the basis of early observation of the differences in tissue damage induced in newborn mice [21]. The first report on oncolytic CVB was published in 1997, which investigated the anti-tumor properties of CVB1 against human colon cancer [22]. Since then, several efforts by different laboratories have been made to assess the oncolytic potential and safety of CVBs targeting various forms of tumor in pre-clinical studies [23,24,25,26,27,28,29,30]. In this review, we summarize the recent progress in developing CVBs, with a focus on CVB3, as an oncolytic virus against different types of cancer. We also discuss the potential barriers and future perspectives for the clinical use of CVBs in cancer therapy.

## 2. Viral Structure and Life-Cycle

The CVBs contain six viral types, named CVB1-CVB6, which are common human pathogens associated with a wide range of diseases from gastrointestinal disorder to aseptic meningitis, myocarditis, and pancreatitis, particularly in infants and children [31,32,33]. The CVBs are small, non-enveloped viruses of approximately 30 nm in diameter [21,34,35]. The viruses contain a positive-sense RNA genome (~7.5 kb) encoding a single, open-reading frame flanked by the 5′ and 3′ untranslated regions (UTRs). The 5’ end of the genome does not contain a cap structure but, instead, is linked to a small viral protein (VPg, also known as 3B, as described below). After receptor binding and uncoating, CVB RNA is internalized and serves as a template for viral protein translation.

The viral genome is translated into a large polyprotein in a cap-independent manner via an IRES located within the 5′UTR [21,34,35]. Subsequently, the polyprotein is processed into individual structural (VP1, VP2, VP3, and VP4) and non-structural proteins (2A, 2B, 2C, 3A, 3B, 3C, and 3D) by virus-encoded proteinases 2A and 3C. Viral RNA-dependent RNA polymerase (RdRp) 3D then synthesizes a negative-strand viral RNA intermediate that serves as a template for the transcription of progeny genomes. Viral replication takes place on virus-modified membranous structures that not only serve as physical scaffolds but also provide favorable lipid compositions for viral assembly and replication [36,37]. The viral proteins 2B, 2C, 3A, and their precursors regulate vesicular transport and the cell permeability required for completion of the viral life-cycle [38]. The 3B protein (VPg), which is linked to the 5′-end of the viral genome, acts as a primer for viral RNA replication [39,40]. The viral capsid is formed by the structural proteins with VP1, VP2, and VP3 exposed on the outer surface of the virion, while VP4 faces the inner surface of the capsid [41]. Finally, viral progeny is released from the infected cells following cell lysis or non-lytically prior to cell rupture via extracellular microvesicles [42,43,44]. In addition to processing viral polyprotein, CVB proteinases cleave multiple host proteins that are essential for the maintenance of cellular architecture, transcription, translation, and anti-viral immunity, thereby contributing to disease progression [45]. The CVB structure and genome are illustrated in Figure 2.

All six types of CVB use the coxsackievirus and adenovirus receptor (CAR) as the main receptor for cell attachment and entry [46,47]. However, some types, such as CVB1, 3, and 5, also use decay-accelerating factor (DAF/CD55) as a co-receptor [48]. CAR is a membrane protein with two Ig-like extracellular domains (D1 and D2) and is expressed in a variety of cell lines, especially in the junction between epithelial cells [49]. CAR binds the surface canyon formed by VP1, VP2 and VP3 of the each protomer of virus capsid to induce the uncoating process [50]. Expression levels of receptors on the surface of tumor cells are an important susceptible factor to oncolytic viruses.

## 3. Features That Make CVBs Attractive Oncolytic Viruses

CVBs have some characteristics that make them promising candidates for oncolytic virotherapy.

First, as RNA viruses, CVBs replicate in the cytoplasm through a negative-sense RNA intermediate, thus avoiding the genotoxicity caused by integration of the viral genome into the host DNA. It has been previously reported that retrovirus-mediated gene therapy of the X-linked severe combined immunodeficiency is associated with the incidence of acute lymphoblastic leukemia, possibly as a result of retrovirus vector integration [52]. In addition, the potential genotoxic side-effects caused by somatic integration remain a risk factor for adenoviral gene therapy [53]. These studies have urged researchers to be cautious about the vector selection and promoted the application of non-integrating vector [54]. Thus, RNA viruses such as CVBs, which do not involve the DNA phase in their life-cycle, seem to serve as ideal candidates for virotherapy.

Second, due to a relatively small RNA genome (~7.5kb), CVBs can be easily manipulated by reverse genetic approaches [21,34]. For example, through reverse transcription of the viral RNA prepared from infected cell culture, viral cDNA can be synthesized and then cloned into a plasmid backbone harboring the T7 or SP6 promoter sequence. After viral genome manipulation, modified viral plasmid can be transcribed into viral RNA with T7 or SP6 RNA polymerase for further use. With the presence of viral IRES in the 5′UTR, capping is unnecessary. For strains of known sequence, direct synthesis of the viral genome is also feasible.

Third, CVBs preferentially replicate and induce lyses in proliferating cells over dormant cells, and their infection largely relies on the activation of oncogenic signaling pathways. It was shown that CVB3 replication is significantly greater in proliferating cells than in G0 or quiescent cells [55]. Evidence has also revealed a crucial role for oncogenic signaling pathways, such as the ERK1/2 and the PI3K pathways, in efficient CVB3 replication [56,57,58]. Aberrant ERK1/2 activation in *KRAS*-mutant cancer cells has also been shown to confer cell susceptibility, at least in part, to CVB3-induced oncolysis [24].

Fourth, CVB infection is controlled by the host’s innate immune response, particularly the type I interferon (IFN-I) signaling [59,60]. Consequently, tumor cells, which commonly have impaired IFN-I response [61], are more sensitive to CVB infection as compared to normal cells. As for CVB3, it has been found that *KRAS* mutation results in compromised IFN-I production in response to viral infection in human lung epithelial cells, contributing to the permissiveness of *KRAS*-mutant lung adenocarcinoma cells to CVB3 infection [24].

Fifth, although CVB infection can be severe in children, leading to myocardial, pancreatic, and neuronal damage, infection in adults is generally asymptomatic or causes only mild flu-like symptoms.

Finally, a large-scale screening of 28 enterovirus strains in vitro has identified CVB2, CVB3, and CVB4 among the most potent enteroviruses that destroy multiple types of cancer cells, including lung, colon, pancreatic, breast, cervical, rhabdomyosarcoma tumor cells [29].

Overall, current evidence suggests that CVBs possess potent and selective oncolytic activities and are ideal candidates for further development into novel oncolytic agents for cancer treatment.

## 4. Opportunities for Oncolytic CVBs in Cancer Therapy

Recent pre-clinical in vitro and in vivo studies have discovered that CVBs have anti-tumor potency targeting various types of cancer, including lung cancer, breast cancer, colorectal cancer, and endometrial cancer.

### 4.1. Lung Cancer

Lung cancer is the most and second-most common cancer type and the leading and secondary leading cause of cancer-related deaths for men and women, respectively [62]. Despite recent advances in immunotherapy, the prognosis of lung cancer, especially *KRAS*-mutant lung adenocarcinoma and small-cell lung cancer (SCLC), remains poor [63,64]. The overall five-year survival rate for all types of lung cancer is only ~19% [65].

The lungs are not normally attacked by CVB3, perhaps due partially to the fact that the expression level of CAR, the primary receptor for CVB3 internalization, is low in normal airway epithelial cells. Of interest, CAR was found to be highly expressed in different lung cancer cells [24,29,66], suggesting the potential application of CVB3 in lung cancer virotherapy. Miyamoto et al. [29] reported that CVB3 (Nancy strain) infects and lyses all lung cancer cell lines investigated, including lung adenocarcinoma (A549, H1299, and LK-87 cells), lung squamous cell carcinoma (EBC-1, QG-56, QG-95, LK-2, and Sq-1 cells), and lung large cell carcinoma (H460 cells) cells, while leaving normal lung fibroblast cells (NHLF and MRC-5 cells) unaffected. Using nude mice bearing A549, EBC-1, and H1299 cell xenografts, they showed that intratumoral administration of CVB3 significantly inhibits growth of the injected as well as non-injected contralateral tumors [29]. Recently, our laboratory also examined the infectivity and cytotoxicity of CVB3 (Nancy strain) in a panel of lung cancer cells and evaluated its efficacy in treating lung cancer in animal models [24,28]. Cell culture studies showed that CVB3 selectively targets Kirsten rat sarcoma viral oncogene homology (*KRAS*)-mutant lung adenocarcinoma (A549, H23, H2030, and HPL1D cell line stably expressing *KRAS^G12V^*) and *TP53/RB1*-mutant SCLC (H524 and H526 cells) cells, with limited impacts on normal lung epithelial cells (primary, BEAS2B, HPL1D, and 1HAEo cells) and epidermal growth factor receptor (*EGFR*)-mutant lung adenocarcinoma (HCC4006, PC9, H3255, H1975, and HPL1D cell line stably expressing *EGFR^L858R^*) [24,28]. Abnormal activation of the extracellular-signal-regulated kinase 1/2 (ERK1/2) pathway and impaired IFN-I innate immune response have been identified as factors determining the susceptibility of *KRAS*-mutant lung adenocarcinoma cells to CVB3-induced cytotoxicity [24]. In vivo investigations using immunocompromised mouse models (NSG and/or NOD-SCID mice) carrying *KRAS*-mutant lung adenocarcinoma or *TP53/RB1*-mutant SCLC xenografts revealed that a single dose of CVB3 through intratumoral or systemic (via intraperitoneal injection) application leads to a marked tumor regression [24,28]. Taken together, these findings suggest that CVB3 is a potent anti-tumor virus against various forms of lung cancer.

### 4.2. Colorectal Cancer

Colorectal cancer is the third most common malignancy and the fourth leading cause of cancer deaths globally, representing an unmet need for effective therapy [67]. An early study reported that CVB1 (Conn-5 strain) infects and lyses human colon cancer cells (SW480 and LIM1215) in a manner depending on the level of epithelial-restricted integrin, αvβ6 [22]. Later, Miyamoto et al. [29] identified that CVB2 (Ohio-1 strain), CVB3 (Nancy strain), and CVB4 (JVB strain) also induce lysis of human colon cancer cells (Caco-2 and DLD-1 cells). Recent studies from the Fechner laboratory explored the oncolytic ability of CVB3 against colorectal cancer [25,68]. It was found that, as compared to the prototype Nancy strain of CVB3 that requires CAR for viral entry, the PD strain of CVB3 that uses heparan sulfate as the primary viral entry receptor more efficiently lyses colorectal cancer cells, including DLD-1, Colo680h, and Colo205 cells, in vitro (although the contribution of minimal expression of CAR in these cells to viral entry cannot be completely ruled out) and inhibits tumor growth in Balb/c nude mice bearing DLD-1 cell xenografts after intratumoral application [68]. The H3 (also known as Woodruff variant) and 31-1-93 (a derivative of PD stain) strains of CVB3 were also shown to be able to potently destroy colorectal cancer cells (DLD-1) in vitro and in nude mice [25,68].

### 4.3. Breast Cancer

Breast cancer is the most common malignancy and leading cause of cancer-related deaths among women [62]. Despite major advances in diagnosis and treatment, triple-negative breast cancer (TNBC), the most aggressive subtype of breast cancer, remains a significant challenge with limited treatment options and a poor prognosis. Miyamoto et al. [29] uncovered a strong lytic ability of CVB2 (Ohio-1 strain), CVB3 (Nancy strain), and CVB4 (JVB strain) towards the MCF7 human breast cancer cell line. A recent follow-up study from the same group revealed that CVB3 (Nancy strain) induces strong oncolytic effects against both human TNBC cells (MDA-MB-231, MDA-MB-468, and MDA-MB-453 cells) and non-TNBC cells (ZR-75-1, SK-BR-3, and MCF7 cells), with no evident harm to human normal mammary epithelial cells (MCF10A cells) [30]. An in vivo animal experiment showed that intratumoral injection of CVB3 significantly limits tumor growth in Balb/c nude mice bearing TNBC MDA-MB-468 cell xenografts [30]. In addition, the adapted variant of CVB6 LEV15 strain was also found to be able to replicate in and lyse MCF7 breast cancer cells in vitro [69]. Intratumoral administration of this variant results in a significant tumor regression in MCF7 cell xenografted nude mice [69].

### 4.4. Other Types of Cancer

It has also been shown that the 2035A strain of CVB3 has an oncolytic activity against human endometrial cancer both in vitro in HEC-1-A, HEC-1-B, and Ishikawa cells and in vivo in nude mice bearing endometrial cancer xenografts, as well as ex vivo using patient-derived endometrial cancer samples [27]. In addition, an anti-tumor ability was also observed for CVB6 (LEV15 strain and its adapted variant) after intratumoral injection into Balb/c nude mice carrying human cervical cancer (C33A cells), prostate cancer (DU145 cells), and rhabdomyosarcoma (RD cells) xenografts [69].

## 5. Challenges and Possible Solutions for the Clinical Use of Oncolytic CVBs

Despite the promising findings about the efficacy of CVBs against different forms of cancer, several barriers remain in developing them for virotherapy, including toxicity, ineffective anti-tumor potency, tumor-specific resistance to CVB-mediated lysis, and delivery inefficacy.

### 5.1. Improvement of Safety and Tumor Specificity

Intratumoral or systemic administration of wildtype (WT) CVBs has been shown to cause injuries to several organs, in particular the heart and pancreas [24,25,26,28]. To reduce the toxicities of CVBs to normal tissues and further enhance their specificity towards tumor cells, several strategies, including the microRNA (miRNA)-based strategy, use of non-pathogenic CVBs, genetical engineering, and selection of non-toxic variants, have been employed (Figure 3):

#### 5.1.1. miRNA-Based Strategy

miRNAs are small (~22 nucleotides in length) non-coding RNAs involved in a broad range of important cellular functions through post-transcriptionally regulating gene expression [70]. miRNAs also play a pivotal role in cancer development and progression. The differential expression profile of miRNAs has been reported in various types of malignancy [71,72]. It is particularly interesting that levels of miRNAs were found to be consistently lower in cancer tissues as compared to normal tissues [73,74]. This trait has been utilized to manipulate CVBs by inserting tumor-suppressive and/or organ-specific miRNA target sequences into the 5′UTR and/or 3′UTR of viral genome to facilitate the degradation of viral RNA in normal but non-cancer tissues, consequently improving the safety profile [75,76].

Jia et al. [26] reported the generation of recombinant CVB3 by introducing target sequences of miR-34a or miR-34c, which are preferentially expressed in normal cells, into the 5′UTR and/or 3′UTR of the viral genome. They found that the CVB3 modified by miR-34a at both the 5′UTR and 3′UTR yields a minimal tissue toxicity while preserving the original oncolytic activity against lung cancer. Our laboratory also exploited miRNA targeting strategy to engineer CVB3 genome to reduce its toxicity [28]. We inserted the target sequences complementary to miR-145/143, which are significantly downregulated in lung cancer cells compared to normal cells, into the 5′UTR of viral genome. An in vivo xenograft mouse study revealed that this modified CVB3 markedly reduces the toxicity of original virus to normal tissues (particularly the heart tissues), while retaining its oncolytic potency against lung cancer [28]. Organ-specific miRNAs have also been utilized for viral genome manipulation. Hazini et al. [25] demonstrated that the modification of CVB3 by inclusion of cardiac-specific miR-1 and pancreatic-specific miR-375 target sequences into its 3′UTR drastically attenuates the unwanted heart and pancreas injuries without compromising its anti-tumor efficacy against colorectal cancer. Similarly, Sagara et al. [30] engineered the CVB3 through inserting target sequences of cardiac-specific miR-1 and pancreatic-specific miR-217 into the 3′UTR of viral genome and showed that the application of the modified virus to nude mice transplanted with human TNBC cells significantly restricts tumor growth without causing apparent side-effects.

#### 5.1.2. Utilization of Non-Pathogenic CVBs

The naturally occurring, non-pathogenic strains of CVBs with evolutionary stability can also be used for oncolytic virotherapy. Several CVB variants have been reported to be non-pathogenic. For instance, the CVB3/0 variant is non-cardiovirulent [77]. The GA strain of CVB3 clinical isolate was also identified to be avirulent when administrated to mice in the absence of evident toxicities to the heart and pancreas [78]. The CVB1N strain of CVB1 is another example of a non-pathogenic variant, which has 23 nucleotide differences compared to the pathogenic strain CVB1Nm [79]. Furthermore, Hazini et al. [68] reported that intratumoral injection of the PD strain of CVB3 efficiently kills colorectal cancer cells in vitro and in vivo with no apparent impacts on normal organs.

#### 5.1.3. Genetic Engineering

Genetic modification of viral genome through mutagenesis provides another useful method for the successful production of viral variants with reduced toxicity. The known sites in the CVB3 genome responsible for the attenuated phenotype of virulent CVB3 have been summarized in a review article of Kim and Nam [80]. For instance, the nucleotide U at site 234 in the 5′UTR of CVB3/0 was identified to be responsible for its non-cardiovirulent phenotype and substitution of U with C restoring the viral toxicity in the heart [81]. Another strategy reported on CVB3 attenuation is the so-called “1-to-stop” method (one substitution can change the Leu/Ser to a stop codon) to add codons that are only one nucleotide substitution away from the stop codons, e.g., Leu (CUA → UUA) and Ser (UCU → UCA) [82]. Such modification attenuates the virus by increasing the rate of nonsense or lethal mutation of viral genome, and thus intervening in the viral protein translation. This 1-to-stop variant replicates significantly less in the heart and pancreas with 100% mouse survival compared to the wide-type (WT) virus with a survival rate less than 30% [82].

The cloning and modification strategy for generating a non-toxic CVB is illustrated in Figure 4.

#### 5.1.4. Selection of Non-Toxic Variants Via Adaptation

Viral adaptation studies have also been used as a strategy to generate viral strains with improved safety profiles [83]. Directed evolution through a serial passage in a target cell line has been demonstrated to be able to obtain CVB strains with attenuated toxicities. For example, an attenuated p14V-1 strain of CVB3 was selected through multiple passages of the cardiovirulent Nancy strain of CVB3 in human dermatofibroblasts [84]. This strain of CVB3 does not cause cardiotoxicity in mice [84,85]. Sequence analysis revealed that the mechanism of attenuation is associated with 23 nucleotide changes compared to the parental cardiovirulent strain [86].

### 5.2. Expansion of Tumor Infectivity Via Adaptation

In addition to being able to improve viral safety as discussed above, serial passages and selections of adapted variants of CVB also allow for broadening viral infectivity to a wider range of malignant cells [83]. Due to genetic plasticity, CVBs can be trained to develop the ability to infect non-permissive cells by introducing adaptive mutations via serial passages in corresponding cells. Examples include the CVB2 Ohio-1 strain [87] and CVB3 RD variant [88], as well as CVB6 LEV15 and Schmitt strains [69,89]. CVB2 and CVB3 do not normally infect rhabdomyosarcoma cells that express the DAF co-receptor but in the absence of CAR. However, it was found that after multiple rounds of selection in the RD rhabdomyosarcoma cell line, these viruses present altered receptor binding preferences and gain the ability to infect these cells using DAF as the viral entry receptor [87,88]. Sequence analysis revealed that one or several mutations in viral capsid regions are likely responsible for the new phenotypes [87,88]. Similarly, the LEV15 strain of CVB6 attains oncolytic capability against previously non-susceptible rhabdomyosarcoma (RD cells) and breast cancer (MCF7 cells) cells after multiple passages in these cells [69]. The selected adaptive strain of CVB3 Schmitt was also shown to acquire the ability to infect human pancreatic duct epithelial cells that are non-permissive to the parental strain [89].

### 5.3. Enhancement of Oncolytic Potency

Oncolytic virus induces the lytic destruction of tumor cells, leading to the release of danger-associated molecular patterns, pathogen-associated molecular patterns, tumor-associated antigens, and cytokines to activate the anti-tumor immune response (Figure 3). However, tumor cells have exerted multiple strategies, such as expressing immunosuppressive molecules, to create an immunosuppressive microenvironment to limit the anti-tumor efficacy of oncolytic virus [90]. Several approaches may be considered to overcome this obstacle, including a combinatorial treatment with cancer immunotherapy and the production of “armed” viruses that encode anti-tumor immunostimulatory molecules such as checkpoint inhibitors, cytokines, or T cell engagers (Figure 3).

Cancer immunotherapy using checkpoint inhibitors, such as anti-programmed cell death-1 (PD-1) and its ligand (PD-L1), has emerged as a new and powerful treatment option for several types of cancer [91]. PD-1 is a T-cell co-inhibitory receptor and blockage of the interaction between PD-1 and PD-L1 leads to increases in T-cell-mediated anti-tumor responses [92,93]. PD-L1 is up-regulated in many forms of cancer cells, contributing significantly to immune escape during tumor development [94]. Upon CVB3 infection, expression of PD-L1 on lymphoid [95] and endothelial cells [96] has also been shown to be upregulated mainly through the IFN-γ signal. These observations suggest that PD-1/PD-L1 may serve as an ideal target to further augment the oncolytic properties of CVBs. Preclinical and early-phase clinical trials have demonstrated that combining oncolytic viruses, including DNA (e.g., HSV, adenovirus, and vaccinia virus) and RNA (e.g., reovirus, VSV, and measles virus) viruses, with PD-1/PD-L1 blockade is more effective than monotherapy, likely through targeting different immunosuppressive pathways [97]. It seems plausible that a combinatorial treatment of oncolytic CVBs and a PD-1/PD-L1 inhibitor could also have an additive anti-tumor effect. Moreover, generation of an armed CVB that expresses an immune checkpoint antibody is another option to promote T-cell-mediated tumor cytotoxicity to further boost host anti-tumor immunity. The CVB genome is relatively small and has limited packaging capacity. The virus tends to get rid of foreign DNA inserts through recombination as replication proceeds. Nevertheless, previous studies on recombinant CVB3 with a GFP insertion (~730 bp) have verified the feasibility of this strategy [44,86]. The advantage of the armed CVBs is to facilitate the local production of anti-PD-1/PD-L1 at the tumor site that can reduce the adverse effects associated with systemic administration [98,99]. In addition to the PD-1/PD-L1 axis, other targets may include anti-tumor cytokines, such as interleukin (IL)-12, IFN-γ, and GM-CSF.

Combination with chimeric antigen receptor (CAR)-T cell therapy has also been tested for several oncolytic viruses, including adenovirus, reovirus, and VSV, and demonstrated improved efficiency in mouse models of solid tumors [100,101,102]. CAR-T cell therapy is based on the technology by which a patient’s T cells are genetically engineered to express modified T cell receptor, so called CAR, to attack cancer cells [103,104]. However, due to the intrinsic ability of tumors to evade immune responses, the application of CAR-T cells in solid tumors has limited benefits [105,106]. As such, oncolytic virus that has different mechanisms of action targeting complement pathways may offer a promising approach to overcome some of the barriers that CAR-T cell therapy encounters in solid tumors [100,102]. Interestingly, there was a recent report showing that IFN-I responses caused by oncolytic VSV induce accelerated apoptosis of the CAR-T cells [107]. Thus, one advantage for CVBs in such a combination treatment might be the impaired host IFN-I innate immunity during CVB infection mainly through the proteolytic activities of virus-encoded proteinases 2A and 3C [108,109], which minimizes the unfavorable impacts of IFN-I on T cells.

In addition to directly boosting anti-tumor immunity of CVBs, optimization of the acidic tumor microenvironment could also increase oncolysis. The acidity within a tumor is a result of high metabolic rate and inadequate perfusion and has been realized as a driving factor in tumorigenesis [110]. It was recently reported that nude mice bearing human lung cancer cell GLC-82, A549, or H460 xenografts treated with a recombinant CVB3 fused with basic peptides exhibit significant tumor regression as compared to mice given non-modified CVB3 [23]. Further investigation showed that this increase in anti-tumor activity is associated with higher pH values within tumors.

### 5.4. Achievement of Efficient Delivery

Delivery of oncolytic virus to tumor cells induces both innate and adaptive immunity, associated with rapid clearance of virus. To maximize the anti-tumor ability, both systemic and local administration may be used [111]. While systemic administration is desirable for cancer therapy, oncolytic HSV-1 is primarily administered through intratumoral injection, even for the first dose, due to pre-existing neutralizing antibody in most individuals. Compared to HSV-1, the prevalence of CVBs appears to be much lower. According to a recent review paper by Brouwer et al. [112], which summarized the data from 153 world-wide studies, the overall prevalence for CVBs is less than ~7.5%, with CVB5 being the highest (~7.5%) and CVB6 the lowest (<0.5%). Despite the relatively low incidence of prior CVB infections, the presence of neutralizing antibodies in a subgroup of people remains a challenge for systemic delivery. In addition, the existence of possible cross-reactive antibodies from other enteroviruses could be another issue [113]. 

The viral genome may be modified to increase its stability in the bloodstream. It was shown that the addition of a CD47 (a “don’t eat me” signal) epitope to the membrane envelop of HSV-1 enables the virus to evade detection by the immune system, thereby enhancing the efficacy of systemic administration [114]. It is speculated that a similar strategy could be applied to the CVB capsid protein to extend viral time in the circulation. Another possible solution to improve the efficacy of systemic administration is the carrier cell-based delivery [115,116]. The carrier cells serve as a delivery cargo that protects the virus from host immune defense. The natural killer cells, NK-92, have recently been tested for the delivery of CVA7 [117]. An in vivo mouse study with glioblastoma xenografts revealed that NK-92-mediated administration of CVA7 increases the delivery efficiency and significantly impedes tumor growth as compared to the direct intravenous injection of viruses, even after only a single injection [117]. Despite limiting the proliferative ability and possible risk of malignant transformation, mesenchymal stem cells as carrier cells have been found to be effective and safe for both the local and systemic delivery of adenovirus in mouse models [118,119]. Future investigations are warranted to assess the efficacy of systemic CVB delivery in immunocompetent animals pre-immunized with WT-CVBs and to explore the approaches to further improve their oncolytic potency.

## 6. Concluding Remarks

As discussed above, CVBs have emerged as promising cancer therapy agents within the past decade. Despite encouraging preclinical data in cells and immunodeficient mice, the research of oncolytic CVBs is still in its early stages and many important questions remain to be answered. Future studies are needed to address the safety and efficacy of oncolytic CVBs in immunocompetent individuals, the exact nature and mechanism that CVBs interact with the host immune system to regulate tumor immune microenvironment and induce systemic anti-tumor immunity, and the best strategies to improve anti-cancer efficacy of CVBs. Future research is also required to identify additional tumor types that CVBs can target and to develop effective delivery approaches for both the primary and metastatic cancer therapy. Nonetheless, there are many opportunities for clinical application of CVBs, possibly together with immune modulators, for the treatment of multiple tumor types. Clinical trials on oncolytic CVBs in the near future are highly anticipated.

## Figures and Tables

**Figure 1 viruses-13-01082-f001:**
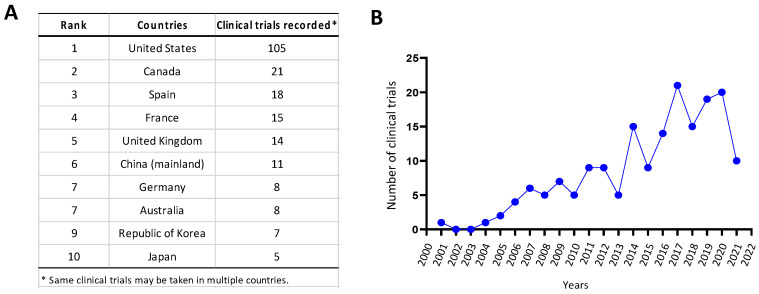
Number of registered clinical trials on oncolytic viral studies from clinicaltrials.gov in top 10 countries (**A**) from 2001 to 2021 (**B**). Note that the data of 2021 only cover clinical studies registered until 31 May 2021. Search keywords ‘oncolytic’ and ‘tumor’ found 182 records in total with five excluded for not involving oncolytic virus.

**Figure 2 viruses-13-01082-f002:**
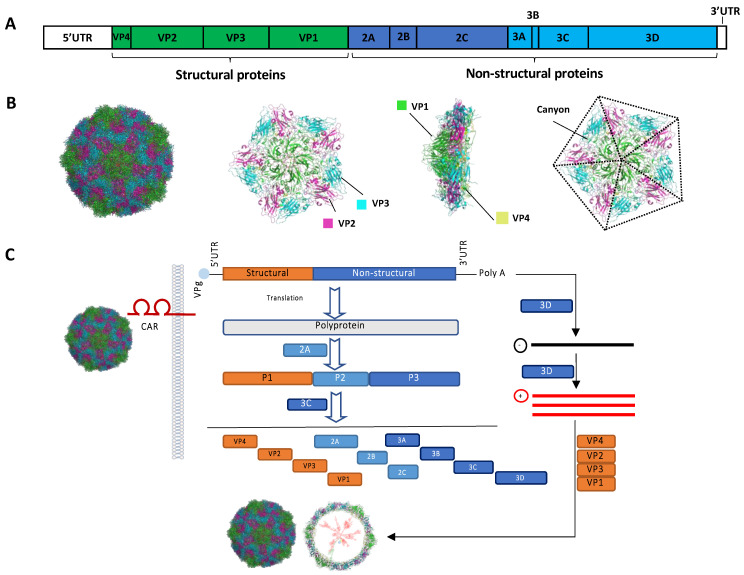
Structure and genome of group B coxsackieviruses. (**A**) Genome structure of coxsackievirus. (**B**) CVB3 structures are generated using the program PyMOL based on the report data by Muchkelbauer et al. [51]. (**C**) Life-cycle of CVB. Following coxsackievirus-adenovirus receptor (CAR)-mediated endocytosis, viral genome is translated into a polyprotein, which is then processed by virus-encoded proteinases 2A and 3C into individual structural and non-structural proteins. Subsequently, cellular machinery is hijacked for viral genome replication through the action of the viral RNA-dependent RNA polymerase 3D. Finally, the viral genome is packaged inside asymmetric protein capsid, followed by viral particle release.

**Figure 3 viruses-13-01082-f003:**
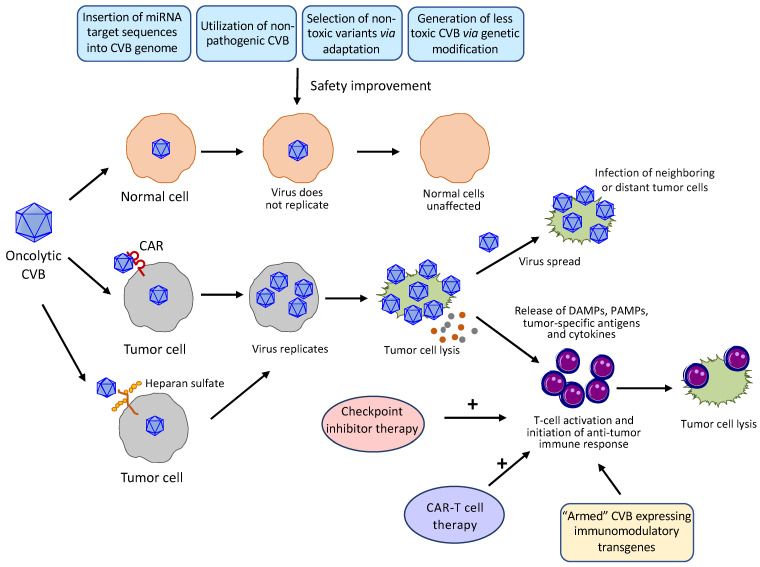
Mechanism of action and strategies applied or proposed to enhance the safety and potency of oncolytic CVBs. CVBs preferentially replicate in and kill tumor cells with limited replication and lysis capacity in healthy cells. Several approaches have been or may be used to further improve the safety profile of CVBs, including insertion of organ-specific/tumor-suppressive miRNA target sequences to viral genome, utilization of non-pathogenic strains of CVBs, selection of non-toxic CVB variants through directed evolution, and production of less toxic CVBs by genetic modification. CVBs enter tumor cells through the coxsackievirus-adenovirus receptor (CAR, primary receptors for all CVBs) and/or heparan sulfate (the PD strain of CVB3). After replication, oncolytic CVBs induce tumor cell lysis, leading to the release of viral progeny that can infect adjacent and distant tumor cells, and the leakage of danger-associated molecular patterns (DAMPs), pathogen-associated molecular patterns (PAMPs), tumor-associated antigens, and cytokines to modulate the tumor microenvironment and activate systemic anti-tumor immune response. A combinational treatment of oncolytic CVB with an immunotherapy, such as checkpoint inhibitor and CAR-T cell therapy, or use of “armed” recombinant CVBs that express immunomodulatory transgenes are expected to yield an additive anti-tumor effect.

**Figure 4 viruses-13-01082-f004:**
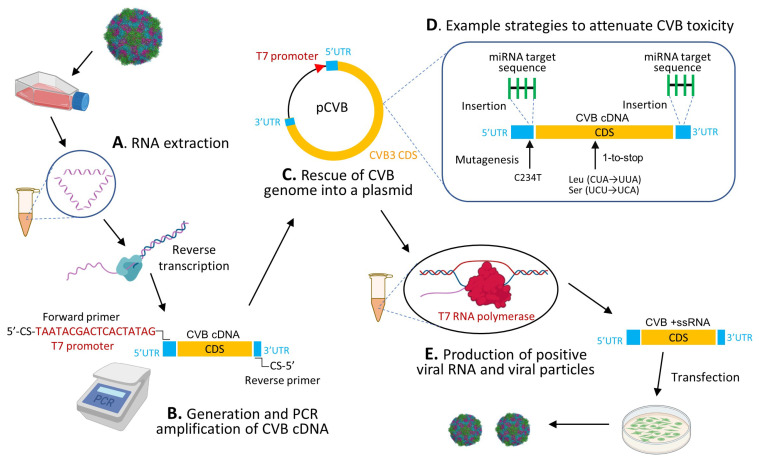
Schematic illustration of the strategy for generating a non-toxic CVB for oncolytic purpose. (**A**) Viral RNA is prepared by RNA extraction from cell cultures. (**B**) Viral cDNA is generated via reverse transcription using poly T primer, followed by PCR amplification with a primer pair containing cloning sites (CS) and T7 promoter sequence (red) in the forward primer. (**C**) Viral cDNA is then rescued into a bacterial plasmid (e.g., pUC18/19). (**D**) Viral genome is modified through insertion of tumor-suppressive and/or organ-specific miRNA target sequences into the 5′UTR and/or 3′UTR of viral genome or by genetic mutagenesis of the viral nucleotides or amino acid codon. (**E**) Finally, the viral RNA is prepared by in vitro transcription with T7 RNA polymerase, and subsequently transfected into CVB susceptible cells to prepare the viral particles for further propagation. CDS, coding sequence.

## Data Availability

Not applicable.

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
