# Peer review of "Development of Group B Coxsackievirus as an Oncolytic Virus: Opportunities and Challenges"

_viruses, 2021, doi:10.3390/v13061082_

Round 1

Reviewer 1 Report

Thank you for nice and prompt paper about the recent progress in this field. What I am really missing is the history of CVBs and particularly the cloning/modification strategies that make the real oncolytic virus vector. Something around this is really needed. Future prospects could be elaborated. Other remarks below:

Abstract: Check the current nomenclature. Human enteroviruses are classified into four species, under which there are named enteroviruses such as coxsackievirus B3. It is erroneous to use term “type B coxsackievirus”. Enterovirus types are “types”, not “serotypes”. Check the recent paper in Archives of Virology https://doi.org/10.1007/s00705-019-04520-6.

Line 55: As per my previous comment, this old classification should be avoided. Although it is correct in what is the outcome in infection into suckling mouse, currently only genetic classification is valid.

Lines 55-60. One should not undermine the work with poliovirus-rhinovirus chimera (PVS-RIPO) and CAVATAK (CVA21). CAVATAK start-up was recently purchased by MSD-Roche with nice sum of money, which could be mentioned as a positive sign of the potential of enteroviruses in oncolytic virotherapy. Few more sentences are needed to emphasize the current knowledge and advances obtained with these viruses, and hopefully there is a niche by which this story focusing only on CVBs is justified.

Line 61. Recent progress? Are there other papers, which mention the old story of CVBs? I think the background of the research is worth mentioning. Even shortly.

Line 65. The introduction could focus on types in EV-B. “Enterovirus B (EVB) species in family Picornaviridae contain X virus types including six coxsackievirus B types (CVB1 - 6) [13,28,29]. There are six serotypes of CVBs, named CVB1-CVB6.

Line 72: VPg aka 3B. Enteroviral proteins are specifically named based on location in the genome as you write in the following sentences.

Line 83. VP4 faces the interior. Not attached.

Figure 2. Nice figures. What is the name of the type? Did you make the images yourself, or describe the source.

One might consider including data on insertion sites, which are discussed in the following sections. While the first part deals with targeting and cytolytic effects of native CVBs, the second part aims to alleviate the problems by modifying the viral genomes. This part is fully missing in the paper.

Line 106. X-linked.

Line 112. One should mention that the downside of the genome size is that it does not tolerate long insert, and is likely to recombine to get rid off them. One could also consider the pathogenicity of CVB3. Disease spectrum was described earlier and included severe diseases, which may affect the use in oncolytic virotherapy.

Line 126. It has been found/detected

Line 170. Heparan sulfate. This may also be an arfifact, since it has been shown that some cell lines express so little CAR that it is not visible in FACS or IFA and hence thought to be CAR(-).

Chapters 3. and 4. should change places: First, describe the potential, second tell about benefits and finally about drawbacks.

Line 201. Figure 3. is placed quite distant from the text. One should list the strategies shortly, and then explain in detail. That is, several strategies including X, Y and Z have been employed (Figure 3):

Line 258: This may also be an arfifact, since it has been shown that RD cell line expresses so little CAR that it is not visible in FACS or IFA and hence thought to be CAR(-). Carson 2007. J. Gen Virol. 88

Line 310. Individuals, not populations.

LIne 311. Could the individual be tested for NAs before administration? This should not be too difficult? How about cross-reacting antibodies from other EVs? Could that be a problem?

Line 321. Just out of curiosity, was this really CVA7? It is really a rare/non-existing type.

Line 346. Many enteroviruses have been passed on to clinical trials. I do not think there is need to emphasize this one. If mentioned, include clinical trial number and not Wuhan or China.

Author Response

We sincerely express our deep gratitude to this reviewer for his/her time and effort in reviewing this manuscript and providing insightful and constructive comments. We have revised the manuscript based on the comments and attached our point-by-point responses in this resubmission.

Reviewer 2 Report

The paper on CVB is an up to date review on the main directions and challenges in developing CVBs as therapeutic platforms for cancer treatment. The article is well written, logically organised and the information is clearly presented. I recommend another read of the article by the authors to rectify minor typos such as double spaces or misplaced words (such as "as" in line 195). 

Round 2

Reviewer 1 Report

Thanks for the corrections and particularly Figure 4. I find the corrections valuable and the paper comprehensive and acceptable for publication.